# The Microbial Genetic Diversity and Succession Associated with Processing Waters at Different Broiler Processing Stages in an Abattoir in Australia

**DOI:** 10.3390/pathogens12030488

**Published:** 2023-03-20

**Authors:** Josphat Njenga Gichure, Ranil Coorey, Patrick Murigu Kamau Njage, Gary A. Dykes, Esther K. Muema, Elna M. Buys

**Affiliations:** 1Department of Consumer and Food Sciences, University of Pretoria, Hatfield 0028, South Africa; elna.buys@up.ac.za; 2Department of Food Science, Nutrition and Technology, South Eastern Kenya University, Kitui P.O. Box 170-90200, Kenya; 3School of Molecular and Life Sciences, Faculty of Science and Engineering, Curtin University, Perth 6845, Australia; r.coorey@curtin.edu.au; 4Division for Epidemiology and Microbial Genomics, National Food Institute, Technical University of Denmark, 2800 Kongens Lyngby, Denmark; panj@food.dtu.dk; 5School of Agriculture and Food Sciences, University of Queensland, St. Lucia 4067, Australia; garydykes66@gmail.com; 6Department of Biochemistry, Genetics and Microbiology, Forestry and Agricultural Biotechnology Institute (FABI), University of Pretoria, Hatfield 0028, South Africa; esther.muema@fabi.up.ac.za

**Keywords:** 16S rRNA amplicon sequencing, chicken, slaughterhouse/abattoir, processing water

## Abstract

The high organic content of abattoir-associated process water provides an alternative for low-cost and non-invasive sample collection. This study investigated the association of microbial diversity from an abattoir processing environment with that of chicken meat. Water samples from scalders, defeathering, evisceration, carcass-washer, chillers, and post-chill carcass rinsate were collected from a large-scale abattoir in Australia. DNA was extracted using the Wizard^®^ Genomic DNA Purification Kit, and the 16S rRNA v3-v4 gene region was sequenced using Illumina MiSeq. The results revealed that the Firmicutes decreased from scalding to evisceration (72.55%) and increased with chilling (23.47%), with the Proteobacteria and Bacteroidota changing inversely. A diverse bacterial community with 24 phyla and 392 genera was recovered from the post-chill chicken, with *Anoxybacillus* (71.84%), *Megamonas* (4.18%), *Gallibacterium* (2.14%), Unclassified Lachnospiraceae (1.87%), and *Lactobacillus* (1.80%) being the abundant genera. The alpha diversity increased from scalding to chilling, while the beta diversity revealed a significant separation of clusters at different processing points (*p* = 0.01). The alpha- and beta-diversity revealed significant contamination during the defeathering, with a redistribution of the bacteria during the chilling. This study concluded that the genetic diversity during the defeathering is strongly associated with the extent of the post-chill contamination, and may be used to indicate the microbial quality of the chicken meat.

## 1. Introduction

Its low-lipid and easily digestible high-quality protein contents make poultry meat the most utilized protein source globally, with the consumption of the chicken meat market having experienced exponential growth to the current 15.1 kg per capita [1]. The microbiome flora on chicken carcasses is associated with survival and persistence during scalding, defeathering, evisceration, and chilling. *Salmonella* spp., pathogenic *Escherichia coli*, *Campylobacter* spp., *Clostridium perfringens*, and *Listeria monocytogenes* are the primary pathogens of concern, with *Lactobacillus* spp., *Lactococcus* spp., *Leuconostoc* spp., and *Pseudomonas* spp. being related to the spoilage of chicken meat [2].

Isolation media coupled with morphology and biochemical testing for confirmation techniques have been the gold standards for microbial detection. The stress factors within bacterial communities impact the accuracy and precision of microbial assessments [3,4]. The carry-over effects from chemical antimicrobials, biofilm formation, and avirulent, viable, non-culturable bacteria understate the level of pathogens that are detected, aggravating the errors [5,6,7]. With only 0.1% of the bacteria detected through culturing, microbiological criteria are shifting to next-generation sequencing, with amplicon profiling and metagenomics being adopted to improve precision for food safety and quality [8,9,10,11,12,13].

Amplicon profiling is achieved through the selective binding of universal primers to the highly conserved and evolutionarily stable 16S rRNA genes, followed by a sequence alignment based on the lengths, positions, and taxonomic discrimination within the hypervariable regions [9,14]. It has been applied to describe the changes in microbial communities, from phylum to genus and species taxa, for an in-depth understanding of the dynamics within microbiome structures and the diversity during the slaughter process [15,16]. Firmicutes, Actinobacteria, Proteobacteria, Cyanobacteria, and Bacteroidetes have been reported as the predominant phyla on chicken carcasses [2,10,17,18,19,20,21]. These microbiome diversity variations depend on age, sex, diet, production system, and chicken health, with about 75% of the caeca microbes being Firmicutes and *Lactobacillus*, *Clostridium*, *Flavonifractor*, *Faecalibacterium,* and *Ruminococcus* being the most abundant genera [2].

Chemical and physical decontamination techniques, coupled with good hygiene practices (GHP) and hazard analysis critical control point (HACCP)-based food safety procedures, have effectively reduced the microbial contamination of chicken carcasses. The high organic content of the associated process water provides a favorable substrate for microbial growth, providing an option for low-cost, non-invasive sample collection, as postulated [22,23]. Understanding the bacterial diversity in this associated processing water will advance the implementation of hygiene and decontamination measures, in order to curb the contamination, re-contamination, and cross-contamination in poultry abattoirs, for safer poultry meats with a more extended shelf life [18,19,24]. There exists a lack of conclusive data on the dynamics of these microbial communities, their survival and persistence, and their genetic diversity at specific processing points. There is also a lack of empirical data on the source attribution of the bacterial genetic diversity on chicken meat using the bacterial community in the processing environment. This research aimed to expand our understanding of the changes in the structure and diversity of the microbiome throughout the slaughter process by profiling the 16S rRNA amplicons. Such information will help to predict the sources and extent of the contamination, cross-contamination, and persistence, and the associated food safety and spoilage risks in commercial, large-scale chicken abattoirs. This study also evaluated the contamination at each processing step to define the probable risk characterization based on the abundance of microorganisms in the processing water.

## 2. Materials and Methods

### 2.1. Sample Collection

The processing water samples were collected from a large-scale slaughterhouse in Australia as follows: (i) inside the scalding tub (*n* = 3), (ii) at the inside–outside-carcass-wash (IOCW) drain (*n* = 3), (iii) inside the chilling unit (*n* = 3), and (iv) at the evisceration drains (*n* = 3). In addition, the researcher collected feather samples (*n* = 3) from the feather-plucking machine and three post-chill broiler chickens (*n* = 9). To account for the variability in contamination that might arise from the differences within flocks, the sampling was done once every month, with samples collected in February, March, and April. The samples were chilled (5–8 °C), transported to the laboratory in sterile plastic containers within 24 h, and analyzed immediately upon arrival.

### 2.2. Sample Preparation

Each carcass was rinsed with 400 mL of buffered peptone water, manually massaged, and placed in an orbital shaker for 2 min. A total of fifty grams of feather samples were weighed and added to 450 mL of buffered peptone water (Oxoid, Basingstoke, UK), vortexed for 60 s, and the rinsate was drained into sterile containers.

#### 2.2.1. Culture-Based Technique

The carcass rinsate and associated process water samples were serially diluted and incubated with the isolation media recommended for *Campylobacter* spp. (ISO10272-2: 2006), *L. monocytogenes* (ISO 11290-1: 2010), *Salmonella* spp. (ISO 6579-1: 2017), and *E. coli* (ISO 3811). For *Campylobacter* spp., 100 µL of the diluent was placed on a modified Charcoal Cefoperazone Deoxycholate Agar (mCCDA) (Oxoid CM0739, Basingstoke, UK) and incubated at 37 °C for 48–72 h under a microaerophilic environment (5% O_2_, 10% CO_2_, and 85% N_2_). For *E. coli*, 100 µL of the diluent was cultured on a Brilliance *E. coli*/coliform Selective Agar (Oxoid CM1046, Basingstoke, UK) and incubated at 37 °C for 24–36 h. For *L. monocytogenes*, Listeria Enrichment Broth (Oxoid CM0862, Basingstoke, UK) was used for its enrichment at 37 °C for 24 h. Afterwards, 100 µL from presumably positive tubes was transferred to the Listeria Selective Agar (Oxoid CM0856, Basingstoke, UK) with a selective supplement (Oxoid SR0140, Basingstoke, UK) and incubated at 37 °C for 24–48 h. For *Salmonella* spp., Mannitol Selenite (M.S.) Broth (Oxoid CM 0399, Basingstoke, UK) with Sodium Biselenite (Oxoid LP0121, Basingstoke, UK) was used for its enrichment at 37 °C for 24–48 h. Another 1ml of the diluent was transferred to Rappaport-Vassiliadis (R.V.) enrichment broth (Oxoid CM0669, Basingstoke, UK) and incubated at 42 °C for 24–48 h. In total, 100 µL from presumably positive M.S. or R.V. broth tubes was spread plated on a Xylose-Lysine-Desoxycholate (XLD) Agar (Oxoid CM0469, Basingstoke, UK) and incubated at 37 °C for 24–48 h. Colony morphology and biochemical tests, namely Gram staining, catalase and oxidase tests, and microscopy, were used to confirm the presumptive colonies for each organism.

#### 2.2.2. 16S rRNA Amplicon Sequencing

DNA extraction was performed using the Wizard^®^ Genomic DNA Purification Kit (Promega, Madison, WI, USA), as per the manufacturer’s protocol, except for the addition of sodium dodecyl sulfate to the nuclei lysis solution. The quality and quantity of the extracted DNA were confirmed using a Qubit Fluorometer (ThermoFisher Scientific, Wilmington, DE, USA) and 1% (*w/v*) agarose gel electrophoresis. 16S rRNA amplicon sequencing was performed using the Illumina MiSeq platform at the Australian Genome Research Facility Ltd. (AGRF) laboratories.

The hypervariable V3-V4 region of the 16S rRNA gene was amplified using a 341F-806R primer set. The forward (CCTAYGGGRBGCASCAG) and reverse (GGACTACNNGGG-TATCTAAT) primer sets each contained Illumina adapter regions (Illumina, Inc., San Diego, CA, USA).

The demultiplexed R1 and R2 sequencing reads, received in fastq format, were analyzed using a Bioconductor (version 3.16) in an R (version 4.2.2) statistical environment. The ShortRead package (version 1.56.1) was used to input the FASTQ files, filter and trim the reads, and generate a quality assessment report [25]. Dada2 (version 1.26.0) truncated the forward and the reverse reads were at 280 and 210, respectively. A statistical denoising and a sample-wise abundance evaluation of the demultiplexed files, in order to remove the substitutions and chimera errors, were performed using the “rdp_gold” database, and the identification of the amplicon sequence variants (ASVs) was also performed using Dada2 [26]. Dada2 used the silva nr99 v138.1 train set database to assign taxonomy up to six taxonomic ranks.

Phyloseq (1.42.0) and Vegan (2.6-4) were used to analyze and graphically present the microbiome data. Alpha diversity was used to explore the microbial richness (using Chao1) and diversity (using Shannon and Simpson metrics). Beta diversity was used to analyze the depth and absolute abundance of the ASV data at the phyla level, using non-metric multidimensional scaling (nMDS) based on the Bray–Curtis and Jaccard dissimilarities. Heatmaps that were generated using Pheatmap (1.0.12) were used to visualize the changes in the structure and abundance of the microbiome at each processing step. Heatmap clustering was performed using Spearman’s rank correlation.

## 3. Results and Discussion

### 3.1. Culture-Dependent Bacteria Assessment

The results of the log-transformed CFU per gram of *E. coli* and *Campylobacter* spp. that were recovered in the processing water samples are presented in Figure 1. *Salmonella* spp. and *L. monocytogenes* were not detected in the enrichment broths. The highest counts of the *E. coli* and *Campylobacter* spp. were from the water samples in the carcass washers and evisceration drains, while the lowest count was in the post-chill carcass rinsate. These findings resonate with previous research and highlight the critical role that scalding, defeathering, and evisceration play in the sharp increase of bacterial contamination and the persistence of bacteria on the carcasses [27].

### 3.2. ASV Abundance

The abundance of the top 10 phyla that were recovered from the processing water and the post-chill carcass rinsate in the 16S rRNA community profiles is presented in Figure 2. A total of 11,042 ASVs were identified from the processing water and post-chill carcass rinsate. From these, 7910 ASVs (7855 bacteria and 55 archaea/parasites) were present in the post-chill carcass rinsate, with Firmicutes (60.02%), Proteobacteria (22.68%), Bacteroidota (11.16%), Actinobacteriota (1.38%), and Desulfobacterota (1.20%) being the abundant bacteria phyla among the 24 recognized phyla. The abundance that was observed on the abundant phylum level resonates with similar studies [20,28]. The differences in the dominant organisms from earlier studies point to differences in abattoir mechanization, levels of personnel hygiene, the uniformity of the chicken slaughter with regard to size and age, and the prevailing season/weather conditions [29]. Halobacterota phylum was the most abundant archaeal phylum, and the results also identified the Parabasalia phylum. The results revealed a decrease in the relative abundance of the Firmicutes during the defeathering (47.89% ↓) and evisceration (24.66% ↓), followed by an increase during the carcass wash (31.74% ↑) and a decrease during the chilling (8.27% ↓), and after that, an increase on the post-chill carcasses (32.60% ↑). An inverse stepwise change in the relative abundance of the Firmicutes with Proteobacteria and Bacteroidota echoes previous studies [18,27,28,30].

Firmicutes in a chicken processing environment are associated with contamination from the crop, stomach, respiratory, reproductive, and cecal matter of a chicken [2]. The abundant Firmicutes in the processing environment were also recovered on the post-chill carcasses. The study revealed a relative abundance of 40.16% in the post-chill carcass rinsate from a total abundance of 875,186 that were recovered from the processing water samples. In total, 127 Firmicutes genera from the initial 203 genera that were identified in the environmental samples survived the processing hurdles. Firmicutes can withstand extreme processing temperatures, chemical and physical decontaminants, and the low oxygen that is associated with scalding, washing, and chilling [19].

On the other hand, the study recovered a relative abundance of only 6.15% of Proteobacteria and Bacteroidota genera in the post-chill carcass rinsate, from an initial total abundance of 536,496 that were recovered from the processing water samples. In total, 77 Proteobacteria and Bacteroidota genera persisted throughout the processing, from an initial 177 genera that were identified in the environmental samples. Our findings confirm earlier findings on the dynamic nature of Proteobacteria and Bacteroidota, and their limited ability to persist and survive in the processing environment [18].

Figure 3 and Table 1 depict the abundance of the main genera in the 16s rRNA community profiles that were recovered from the post-chill whole carcass rinsate samples, with an abundance of different processing points. The 7855 ASVs represented 473 bacteria genera (392 identified and 81 unclassified genera). The abundant genera within the Firmicutes in this study were *Anoxybacillus* (38.84%), *Megamonas* (5.57%), *Lactobacillus* (3.89%), Unclassified Lachnospiraceae (2.57%), and *Tepidimicrobium* (1.75%). *Dechloromonas* (5.54%), Unclassified Enterobacteriaceae (2.90%), *Yersinia* (2.74%), *Gallibacterium* (2.51%), and *Acinetobacter* (1.20%) were the most abundant genera within the Proteobacteria. Unclassified Prevotellaceae (1.53%), *Bacteroides* (1.04%), and *Williamwhitmania* (0.68%) genera were the most abundant genera within the Bacteroidota.

*Salmonella* and *Listeria* genera were not recovered in the 16S rRNA amplicons and collaborated with the culture-based detection data that were presented in Section 3.1. Furthermore, *Campylobacter* was abundant in the processing water samples, including in the chillers, which points to the presence of non-culturable bacteria during the chilling [5,31]. Unclassified Enterobacteriaceae revealed similar trends to the *E. coli* counts at the different points throughout the slaughter process, with a total abundance of 62,995. The low abundance of *Escherichia-Shigella* could be due to the potential underrepresentation of this genus in the reference silva database [32].

### 3.3. Cluster Analysis

The cluster analysis results of the log10 transformed community abundance, which was used to associate the contamination in the processing water from the different processing points throughout the slaughter operation to those in the chilled whole carcass rinsate, are shown in Figure 4A. The Spearman’s rank correlation that is presented in Figure 4B confirms that the community at the defeathering and chilling had the highest association with those in the post-chill carcass rinsate. *Anoxybacillus*, *Tepidimicrobium*, *Tepidiphilus*, *Thermus,* and *Corynebacterium* were clustered to represent the selective heat-resistant mesophilic organisms that survive scalding and chilling temperatures. Unclassified Enterobacteriaceae, *Lactobacillus*, *Gallibacterium*, *Streptococcus*, *Acinetobacter*, *Enterococcus*, Unclassified Veillonellacaea, and *Pseudomonas* were clustered to represent the psychotropic fermentative bacteria originating from the gastral intestinal with the ability to persist during washing and chilling. *Staphylococcus*, Unclassified Comamonadaceae, *Moraxella*, *Fusobacterium*, *Acetoanaerobium,* and Unclassified Neisseriacaea survived during the chilling, which points to possible cross-contamination in the chill tanks.

The evisceration drain samples were the most distant from the post-chill carcasses; Proteobacteria was the dominant phylum, with *Dechloromonas* (28.63%), *Yersinia* (14.18%), and *Gallibacterium* (4.98%) being the abundant genera. In contrast to Hauge et al. (2023), this study revealed that Gram-negative Proteobacteria, such as Unclassified Enterobacteriaceae, *Gallibacterium*, *Acinetobacter,* and *Moraxella,* and Gram-negative Bacteriodota, such as *Prevotella* and *Bacteroides*, were able to survive and persist within the processing environment [33]. The fermentative anaerobic and microaerophilic taxa that are associated with the chicken gastral-intestinal persisted throughout the processing line, which confirms the earlier postulation [2,21].

### 3.4. Alpha Diversity

From the statistical evaluation of the alpha diversity, the research observed highly divergent bacterial communities at the different points in the abattoir, based on the richness, chao1, evenness, and Shannon matrices (Figure 5). Most of te samples had richness scores ranging between 300 and 1600 ASVs. Similar trends were also observed from chao1, implying that the sequencing depth was adequate to account for all the diversity in the sampled environments. The intra-variability of the samples collected at the scalder drain and post-chill carcass rinsate, with the rest of the processing water samples, was observed from the evenness and Shannon indices.

A one-factor ANOVA test on the Shannon index revealed a significant effect of the sampling point (*p* < 0.001), while the sampling month had no significant impact on the diversity (*p* = 0.937). Similar findings were observed from the Kruskal–Wallis rank sum test on the Shannon index. The sampling month had a Kruskal–Wallis chi-squared of 0.348 with a *p*-value = 0.840, while the processing water samples had a Kruskal–Wallis chi-squared of 11.867 with a *p*-value = 0.037.

Alpha diversity indices were used to indicate the potential hotspots for contamination, with their results revealing higher risks of the contamination of chicken carcasses from the feathers and gastrointestinal content during defeathering, with a potential cross-contamination during chilling. The effects on the alpha diversity of genetically distinct taxonomic ASVs resonate with previous studies, where a steady increase in the richness and evenness was observed from scalding to chilling, with a slight decrease at post-chill [18,28]. The close resemblance of the alpha diversity between the microbial community that were recovered in the post-chill carcass rinsate with those from the defeathering tub revealed their persistence on chicken carcasses, and their ability to resist washing and chilling, as previously described [27]. Similarly, low temperatures and chemical decontamination during the chilling decreased the richness but did not significantly impact the Shannon indices of the chiller waters [30]. The season has been reported to significantly influence the alpha diversity, with more complex bacterial community structures being recovered during the washing and chilling water in the summer [29].

### 3.5. Beta Diversity

Figure 6A shows the variability between the samples using a two-dimension principal coordinate analysis (PCoA) plot. The PCoA plot revealed that 60% of the total variance between the samples was accounted for by the sampling time (month) and sample type. Both axes differentiate the samples that were collected from the different processing points in different months. The intra-variability between the samples collected in March was much lower than those from February and April. In total, three distinct groups were observed, which revealed that the intra-variability within the sampling points was higher than the inter-variability between the months.

Figure 6B visualizes the differences in the bacterial communities that were recovered from the water samples that were collected at the different points in the abattoir in different months, using non-metric multidimensional scaling (nMDS). A clear separation of clusters was observed from the samples that were collected at various points, and, to a lesser extent, from the samples that were collected in different months. A permutational multivariate analysis of variance (PERMANOVA) of the absolute abundance of the ASVs based on Bray’s model revealed a significant difference between the processing water samples that were collected at the different points in the abattoir (*p* = 0.010). At the same time, the sampling month did not significantly influence the β-diversity (*p* = 0.410). This was confirmed by the permutation test using Jaccard’s model, which similarly revealed a significant difference for the samples that were collected at the different points (*p* = 0.001), with the month of the sampling being insignificant on the β-diversity (*p* = 0.344).

From both the PCoA and nMDS matrices, the β-diversity revealed a clear separation of clusters from the samples that were collected at the different points, and, to a lesser extent, from the samples that were collected in different months. This confirmed that biases other than the sampling points do not significantly influence both the abundance-weighted and unweighted matrices. The beta diversity, using nMDS, has previously revealed significant dissimilarities between the diversity of the bacterial communities that were collected from the different processing water samples [28,29].

## 4. Conclusions and Recommendations

The recovery and enumeration of *E. coli* and *Campylobacter* spp. in processing water during scalding, defeathering, evisceration, carcass-wash, and chilling indicate the contamination of chicken meat. The 16S rRNA amplicon sequencing application provides a food safety surveillance tool that is capable of detecting non-culturable and injured bacteria at different points throughout the slaughter process. The microbial community that is present during defeathering has the potential to estimate the extent of the microbial safety and shelf life of chicken meat, although further studies are needed to confirm this. The chilling waters provide a platform to assess the cross-contamination and redistribution of microorganisms during immersion chilling. The diversity of the genetically distinct taxonomic ASVs increased steadily from scalding through defeathering, evisceration, and washing to chilling, and these steps acted as hotspots of contamination and cross-contamination.

## Figures and Tables

**Figure 1 pathogens-12-00488-f001:**
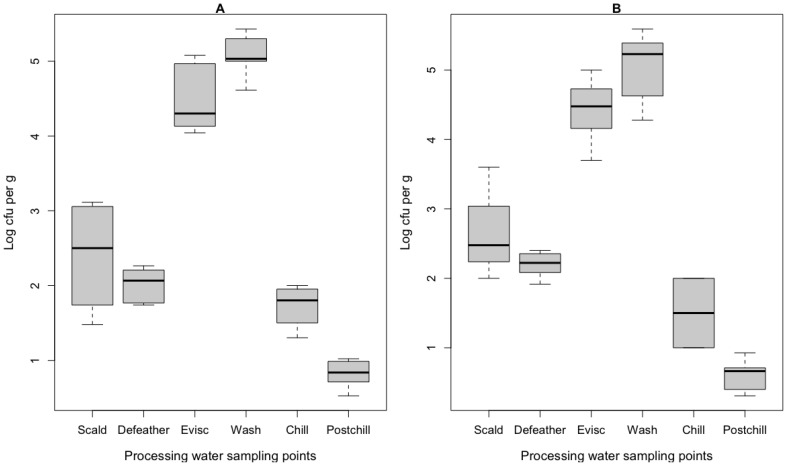
Boxplot of the log CFU per g (*n* = 3) from processing water collected from the different points throughout the broiler slaughter operation; (**A**) *Campylobacter* spp.; and (**B**) *E. coli*. Evisc = processing water sampled in the evisceration. The boxes represent the median, quartile range, and maximum/minimum values.

**Figure 2 pathogens-12-00488-f002:**
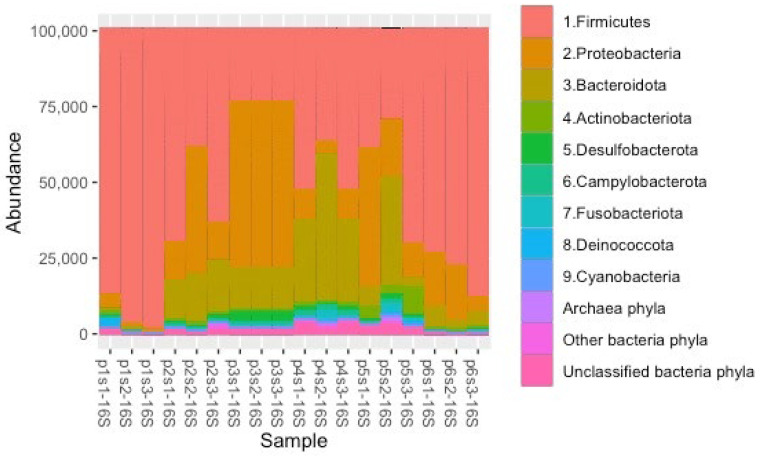
Relative abundance of main phyla in the 16s rRNA community profile, identified in the processing waters and carcass rinsate samples. P1 = scalders water, P2 = feather drain water, P3 = evisceration drain water, P4 = carcass-washers water, P5 = chillers water, and P6 = whole carcass rinsate of chilled carcasses. S1 = samples collected in February, S2 = samples collected in March, and S3 = samples collected in April.

**Figure 3 pathogens-12-00488-f003:**
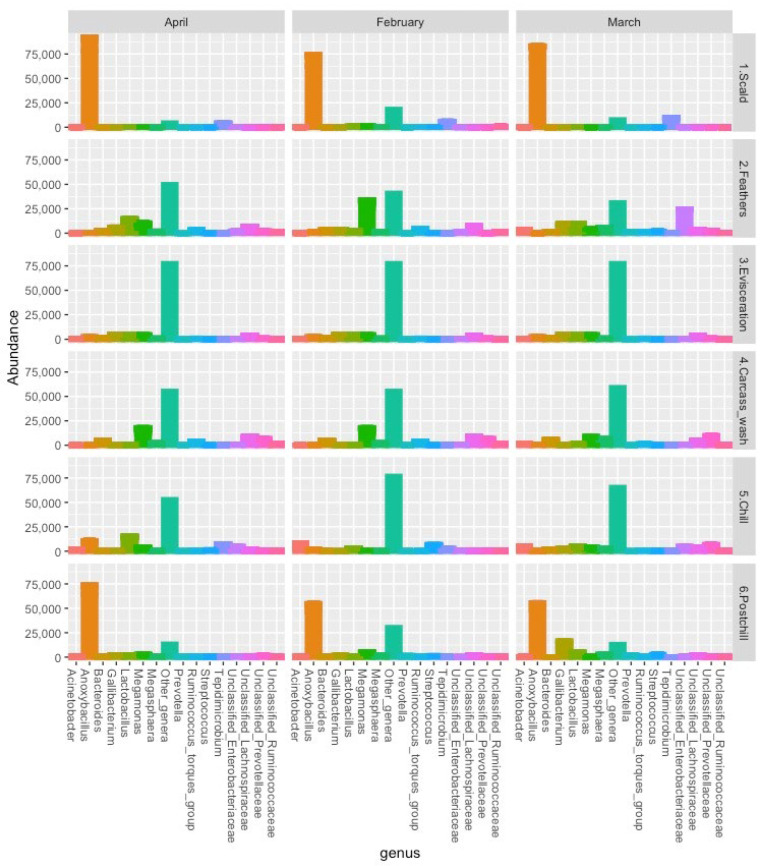
The abundance of the main genera recovered from processing water samples from different processing positions in the chicken abattoir in the 16s rRNA community profiles.

**Figure 4 pathogens-12-00488-f004:**
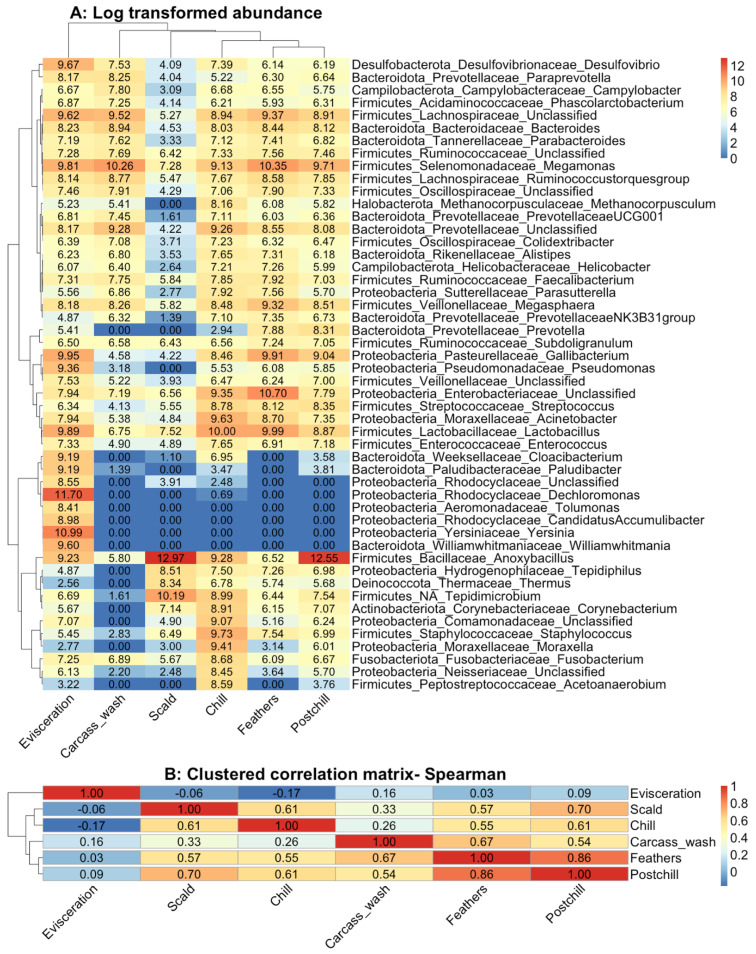
(**A**) The log _10_ transformed microbial community heatmap of the top 50 abundant genera recovered from processing water sampled at different processing points (*n* = 3). Dendrograms represent hierarchical cluster analysis grouping of the genera; and (**B**) Spearman’s correlation of the microbial communities sampled from different sources.

**Figure 5 pathogens-12-00488-f005:**
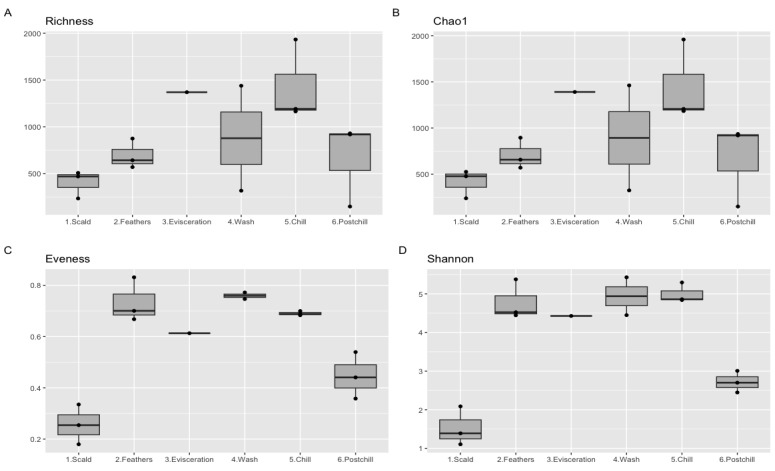
Box plot to represent the alpha diversity measured using (**A**) richness; (**B**) chao1, (**C**) evenness, and (**D**) Shannon metrics. The boxes represent the median and the 25 and 75% interquartile range.

**Figure 6 pathogens-12-00488-f006:**
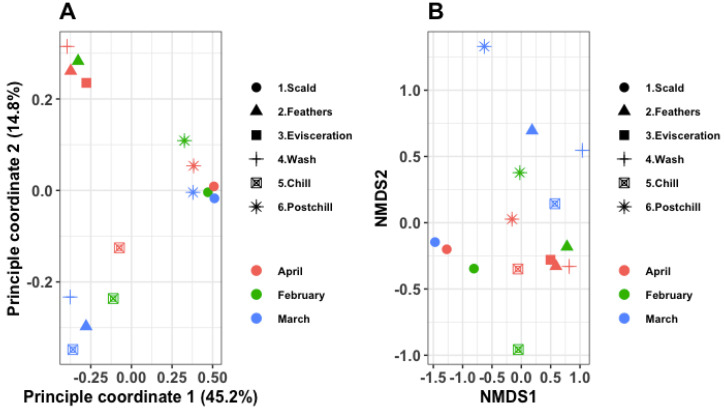
(**A**) Two-dimensional PCoA plot highlighting the samples’ variance; and (**B**) dissimilarities between bacterial communities at different points throughout the slaughter process, using non-metric multidimensional scaling (nMDS) of absolute abundance ASVs based on the square root transformation of Bray–Curtis.

**Table 1 pathogens-12-00488-t001:** The genera exceeding the mean abundance of 1000 recovered from the chilled whole carcass rinsate, with the respective abundance recovered from the processing water samples.

Genus	Scald	Feathers	Evisceration	Carcass wash	Chill	Post-Chill
*Anoxybacillus*	429,789	678	10,239	328	10,714	283,148
*Megamonas*	1444	31,374	18,165	28,700	9187	16,475
*Gallibacterium*	67	20,199	20,904	97	4705	8444
Unclassified Lachnospiraceae	193	11,735	15,081	13,627	7655	7382
*Lactobacillus*	1842	21,819	19,797	855	22,119	7101
*Megasphaera*	337	11,142	3585	3873	4838	4948
*Streptococcus*	256	3358	564	61	6479	4211
*Prevotella*	0	2648	222	0	18	4074
*Bacteroides*	92	4622	3738	7598	3068	3352
Unclassified Prevotellaceae	67	5180	3531	10,735	10,526	3216
*Ruminococcus* (torques group)	236	5327	3435	6453	2152	2553
Unclassified Enterobacteriaceae	704	44,191	2796	1325	11,554	2425
*Tepidimicrobium*	26,575	627	801	4	7988	1886
Unclassified Ruminococcaceae	611	1916	1455	2191	1528	1737
*Acinetobacter*	125	6000	2811	216	15,285	1552
Unclassified Oscillospiraceae	72	2702	1728	2717	1165	1522
*Enterococcus*	132	1005	1527	133	2094	1310
*Catenibacterium*	0	1249	105	0	9	1216
Unclassified Moraxellaceae	61	739	102	36	1729	1184
*Corynebacterium*	1256	468	288	0	7427	1181
*Subdoligranulum*	617	1387	666	722	704	1152
*Faecalibacterium*	344	2757	1488	2325	2572	1126
Unclassified Veillonellaceae	50	510	1866	184	642	1097
*Staphylococcus*	659	1890	231	16	16,877	1090
*Tepidiphilus*	4960	1418	129	0	1806	1078

The genera have been ranked based on the abundance recovered on the post-chill carcass.

## Data Availability

The data presented in this study are available upon request from the corresponding author.

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
