# Peer review of "The Microbial Genetic Diversity and Succession Associated with Processing Waters at Different Broiler Processing Stages in an Abattoir in Australia"

_pathogens, 2023, doi:10.3390/pathogens12030488_

Round 1

Reviewer 1 Report

Very specific research for the segment of genetic microbiology. However, it does not fail to involve all the care that must be taken during the entire process of slaughtering chickens in order to have healthy meat.

Author Response

I take this opportunity to appreciate your time and effort in reviewing the article and for the valuable compliment on the manuscript. 

Reviewer 2 Report

The subject of the impact of microbiome of poultry processing environment on bacterial pathogen persistence is a very important and topical issue. In summary, the manuscript is well written with detailed background information and literature to support the rationale and significance of the experiment. Some minor edits are noted below:

Line 294: change Figure 52 to Figure 5.

Line 315: remove the hyphen in collect-ed.

Line 320: remove the hyphen in in-fluence.

Author Response

I take this opportunity to appreciate your time and effort in reviewing the article and for the valuable compliment on the manuscript. 

As suggested, the revised manuscript has been checked by a native English-speaking colleague and revisions done using track changes.

Point 1: Line 294: change Figure 52 to Figure 5.

Line 315: remove the hyphen in collect-ed.

Line 320: remove the hyphen in in-fluence.

Response 1: These have been edited in the revised manuscript.

Reviewer 3 Report

Attached

Author Response

We take this opportunity to pass our appreciation for your time and effort in reviewing the manuscript and for the valuable comments on improving it. Kindly find our detailed responses attached.
